# Cancer Tissue Classification Using Supervised Machine Learning Applied to MALDI Mass Spectrometry Imaging

**DOI:** 10.3390/cancers13215388

**Published:** 2021-10-27

**Authors:** Paul Mittal, Mark R. Condina, Manuela Klingler-Hoffmann, Gurjeet Kaur, Martin K. Oehler, Oliver M. Sieber, Michelle Palmieri, Stefan Kommoss, Sara Brucker, Mark D. McDonnell, Peter Hoffmann

**Affiliations:** 1Future Industries Institute, University of South Australia, Mawson Lakes 5095, Australia; Parul.Mittal@unisa.edu.au (P.M.); Mark.Condina@unisa.edu.au (M.R.C.); Manuela.Klinger-Hoffmann@unisa.edu.au (M.K.-H.); 2Clinical & Health Sciences, University of South Australia, Adelaide 5001, Australia; 3Institute for Research in Molecular Medicine, Universiti Sains Malaysia, Minden Penang 11800, Pulau Pinang, Malaysia; gurjeet@usm.my; 4Department of Gynaecological Oncology, Royal Adelaide Hospital, North Terrace, Adelaide 5000, Australia; martin.oehler@adelaide.edu.au; 5Robinson Research Institute, Discipline of Obstetrics and Gynaecology, Adelaide Medical School, University of Adelaide, Adelaide 5005, Australia; 6Personalised Oncology Division, The Walter and Eliza Hall Institute of Medial Research, Parkville 3052, Australia; sieber.o@wehi.edu.au (O.M.S.); palmieri.m@wehi.edu.au (M.P.); 7Department of Medical Biology, The University of Melbourne, Parkville 3052, Australia; 8Department of Surgery, The University of Melbourne, Parkville 3050, Australia; 9Department of Biochemistry and Molecular Biology, Monash University, Clayton, Melbourne 3800, Australia; 10Department of Women’s Health, Tübingen University Hospital, Calwerstr. 7, 72076 Tübingen, Germany; Stefan.kommoss@med.uni-tuebingen.de (S.K.); sara.brucker@med.uni-tuebingen.de (S.B.); 11Computational Learning Systems Laboratory, UniSA STEM, University of South Australia, Mawson Lakes 5095, Australia

**Keywords:** colorectal cancer (CRC), endometrial cancer (EC), lymph node metastasis (LNM), machine learning (ML), matrix assisted laser desorption/ionization mass spectrometry imaging (MALDI MSI)

## Abstract

**Simple Summary:**

Classic histopathological examination of tissues remains the mainstay for cancer diagnosis and staging. However, in some cases histopathologic analysis yields ambiguous results, leading to inconclusive disease classification. We set out to explore the diagnostic potential of mass spectrometry-based imaging for tumour classification based on proteomic fingerprints. Combining mass spectrometry with supervised machine learning, we were able to distinguish colorectal tumor from normal tissue with an overall accuracy of 98%. In addition, this approach was able to predict the presence of lymph node metastasis in primary tumour of endometrial cancer with an overall accuracy of 80%. These results highlight the potential of this technology to determine the optimal treatment for cancer patients to reduce morbidity and improve patients’ outcomes.

**Abstract:**

Matrix assisted laser desorption/ionization mass spectrometry imaging (MALDI MSI) can determine the spatial distribution of analytes such as protein distributions in a tissue section according to their mass-to-charge ratio. Here, we explored the clinical potential of machine learning (ML) applied to MALDI MSI data for cancer diagnostic classification using tissue microarrays (TMAs) on 302 colorectal (CRC) and 257 endometrial cancer (EC)) patients. ML based on deep neural networks discriminated colorectal tumour from normal tissue with an overall accuracy of 98% in balanced cross-validation (98.2% sensitivity and 98.6% specificity). Moreover, our machine learning approach predicted the presence of lymph node metastasis (LNM) for primary tumours of EC with an accuracy of 80% (90% sensitivity and 69% specificity). Our results demonstrate the capability of MALDI MSI for complementing classic histopathological examination for cancer diagnostic applications.

## 1. Introduction

Tissue examination by histopathology is the established ‘gold standard’ for cancer diagnosis and staging in clinical laboratories. However, in some cases histopathologic analysis yields ambiguous results leading to inconclusive disease classification. The standard workflow in pathology laboratories is sectioning the tumour tissue and staining with hematoxylin and eosin (H&E) stain before visual inspection by a trained pathologist. Additional information which might be relevant for staging or therapy decision can be provided by immunohistochemistry (IHC) staining if appropriate biomarkers have been identified and specific antibodies are available. For example, IHC assessment of estrogen receptor (ER), progesterone receptor (PR) and HER2 expression defines major subtypes of breast cancer and is predictive of the risk of recurrence and overall survival (reviewed in [1]. Matrix-assisted laser desorption/ionization mass spectrometry imaging (MALDI MSI) is becoming increasingly popular in both medical and basic research for discovery of detailed histo-molecular signatures of tissues, such as proteins, peptides, glycans, or metabolites, without the need of tagging or target specific antibodies [2]. MALDI MSI has been successfully used to classify tumours to provide differential diagnosis, to predict disease prognosis, and to predict response to therapy (reviewed in [3]. The ability of MALDI MSI to profile biomolecule(s) with differential spatial localization enables determination of tissue identity, analysis of heterogeneity and studies of tumour-stroma interrelationships [4].

Two broad categories of ML can be applied to MALDI MSI data, supervised and unsupervised learning [5,6,7,8] Generally, in any application of ML, the choice between supervised and unsupervised is determined by the data that is available. Unsupervised learning is suitable when samples available for training a ML model are not tagged with “ground-truth labels” which are a requisite for supervised learning. Such labels might not be available for many reasons, ranging from the purpose of applying ML to the case where “ground-truth” is too expensive or difficult to acquire in sufficient volume for applying supervised learning. In the area of MALDI MSI, adoption of unsupervised ML approaches for data processing, pattern recognition and exploration has been successfully applied to elucidate molecular signatures, as reviewed in [6]. Examples of prior applications of unsupervised learning to MALDI data include the characterization of epithelial ovarian cancer histotypes [9], white/grey brain tissue [10], renal cancer [7], pulmonary arterial hypertension (PAH) [11].

Supervised ML approaches have been explored in the context of clinical diagnostic applications, such as discrimination of tumour from non-tumour tissues, with a potential to automate routine pathology [7,12]. Moreover, adoption of MALDI MSI as a routine platform holds promise for providing additional molecular biomarker information that cannot be obtained using traditional histopathological inspection. For example, deep neural networks have been applied to MALDI MSI data to classify two lung tumor subtypes [7], and non-negative matrix factorization methods to discriminate lung and pancreas tumors [12]. Other examples include classification of six common cancer types [13] and discrimination of breast and ovarian cancer [14] or metastatic breast and pancreatic cancer [15]. Our previous work has adopted supervised ML classification methods to predict the metastatic status of primary EC tissues, achieving 88% accuracy, although sample size was limited [16]. Here, we extended on previous supervised ML approaches to examine the utility of deep neural networks, applied to large MALDI MSI datasets from CRC and EC (>300 patients for each cohort), for two alternative approaches to binary classification, sample classification by spectra and sample classification by tissue group. To pursue this evaluation, we created pathologist-labelled datasets to define high-quality ”ground-truth” for defined classification problems, tumour vs. normal tissue for CRC and presence and absence of LNM in primary tumours of EC. For CRC, a total of 15 TMAs were analyzed by tryptic peptide MALDI MSI consisting of 564 tissue groups (286 normal, 278 tumour) from 302 unique patients. For EC, three TMAs were analyzed consisting of 303 patient tissue samples, of which 43 patients were pathologist-classified as with LNM and 214 as without LNM.

For sample classification by spectra, all spectra were treated as independent variables, such that no contextual information was available for training of the ML models. For sample classification by tissue group, sets of spectra from all cores of a single tissue type were jointly considered to derive group prediction models. Model performance was assessed by cross-validation procedures. Given the datasets available to us, we set out with two ML aims, as articulated in the following:

Aim 1: Use Machine Learning to Classify Spectra

“One spectrum is one sample”: The aim here was to design and validate a binary classifier for which category individual spectra belong to (normal or tumour tissue cores for CRC and tissue cores with or without LNM for EC). All spectra were therefore treated as independent samples, such that no contextual information was available for ML models to learn from. As we will show, the designed ML model correctly classified ~98% of all CRC spectra as derived from normal or tumorous tissue and ~80% of all EC spectra according to their metastasis stage.

Aim 2: Use Machine Learning to Classify Tissue Groups

“One tissue group is one sample”: Given the set of all spectra from all cores of a single tissue type (e.g., normal or tumour for CRC) from one patient, the aim here was to design and validate a method for fusing all the predictions for single spectra in a tissue group to then classify the group. For our EC data this is a “one patient is one sample” scenario, whilst for our CRC cohort most patients have data for each tissue group (i.e., normal and tumour). As we will show, for CRC out of 564 tissue groups, 556 were correctly classified (cancer or normal) in our machine-learning cross-validation process, while for EC, 159 out of 200 samples were correctly classified (LNM positive or LNM negative) in 10-fold cross validation splits (each consisting of 10 positive and 10 negative samples). The code we used to implement machine learning models and data preparation for these models is available at the following GitHub repository: https://github.com/McDonnell-Lab/MALDI-UNISA-ML, accessed on 24 October 2021.

## 2. Materials and Methods

### 2.1. Tissue Samples and Tissue Microarray

#### 2.1.1. Colorectal Cancer

For CRC, a total of 15 formalin-fixed paraffin-embedded (FFPE) TMAs were examined, representing tumor and normal tissues from 302 stage I-VI patients recruited at the Royal Melbourne Hospital, Melbourne Private Hospital and Western Hospital Footscray in Melbourne, Australia. All patients gave informed consent, and the study was approved by the relevant hospital and institutional ethics committee of the Walter and Eliza Hall Institute (WEHI HREC 12/19). Each CRC TMA was designed in a grid with up to 132 tissue cores from 22 patients arrayed in a grid of 11 rows and 12 columns. For each patient, there were typically four ‘tumour’ tissue samples and two ‘normal’ samples, arranged as shown in Figure 1A. TMA blocks were sectioned at 6 µm thickness and mounted onto conductive indium tin oxide slides (ITO, Bruker Daltonik, Bremen, Germany) for MALDI MSI, and super-frost slides (Thermo Scientific, Germany) for H&E staining. Diagnosis on H&E-stained TMAs was confirmed by an experienced pathologist and each tissue core was annotated encircling tumour regions in black and normal regions in green as shown in Figure 1B.

#### 2.1.2. Endometrial Cancer

For EC analysis, three FFPE TMAs were analyzed consisting of tissue cores from 302 unique patients. The pathologist labels for these patients included 43 patients of primary tumour with LNM, 214 patients with no LNM, and 45 patients that were unclassified/not tested. Each patient had either two or four tissue cores on the TMAs, an example layout is shown as Appendix A. There were 786 total cores on the 3 TMAs. The study was approved by ethics committee of the University of Tübingen (488/20218BO2) and informed consent was given by all patients prior to participating in the study. As with the CRC TMAs, an experienced pathologist annotated each tissue core encircling regions with LNM in red while without in blue as shown in Figure 2.

### 2.2. MALDI MSI Sample Preparation

Tryptic peptide MALDI MSI was performed on sections cut from CRC and EC FFPE TMAs as per the protocol of Gustafsson et al. [17] with minor modifications. Briefly, the tissue sections were heated at 60 °C for 60 min, de-waxed with xylene (Chem- Supply, Australia) twice for 5 min, re-hydrated through graded level of ethanol (Merck, Australia) for 2 min each (100%, 90% 80% and 70%, *v*/*v*), followed by 15 min incubation with 10 mM ammonium bicarbonate (Merck, UK), twice. Heat induced citric acid antigen retrieval (10 mM citric acid, pH 6, Sigma-Aldrich, Japan) was performed in a microwave (LG 700 W MS19496, LG, China) for 10 min at 98 °C and subsequently onto a heating block at 98 °C for 30 min. Slides were cooled down at room temperature followed by two 60 s incubation in10 mM ammonium bicarbonate and drying at room temperature for 10 min. Teach points were marked on the slides using a whitener and optical images were recorded using a flattop scanner (CanoScan 5600F, Canon, Thailand).

Next, 20 µg of trypsin gold (Promega, USA) in 25 mM ammonium bicarbonate (Merck, UK) with 10% acetonitrile (*v*/*v*, Merck, Germany) was deposited onto the slide using ImagePrep (Bruker Daltonik, Germany) in 30 cycles with a fix nebulization time of 1.2 s and drying time of 15 s followed by incubation at 37 °C for 2 h in a humidified chamber. A mixture of internal calibrants [18] was sprayed using the same method on the ImagePrep as of trypsin. Next, 7 mg/mL of alpha-cyano-4-hydroxycinnamic acid (HCCA, Bruker Daltonik, Germany) in 50% acetonitrile (*v*/*v*, Merck, Germany) with 0.2% trifluoroacetic acid (*v*/*v*, Merck, Germany) was deposited onto the slide using an optimised sensor-controlled nebulization matrix deposition method on the ImagePrep system. Re-crystallization of the matrix was performed at 85 °C for 1.5 min in a humidified chamber as described in [19].

### 2.3. MALDI MSI Data Acquisition

MALDI MSI data was acquired on an ultraflextreme MALDI TOF/TOF mass spectrometer equipped with smartbeam-II laser. (Bruker Daltonik, Germany) controlled by flexControl v3.4 and flexImaging v4.1 software (Bruker Daltonik, Germany). The instrument was operated in positive reflectron mode over the mass range of *m*/*z* 800–4500 Da at a spatial resolution of 70 µm with a laser repetition rate of 2 kHz. A total of 1000 laser shots were applied in each sample spot. Raw datasets were recalibrated in flex Analysis v3.4 (Bruker Daltonik, Germany) using the sprayed internal calibrants (angiotensin, Glu-Fib, dynorphin and ACTH 1-24) in quadratic mode with peak alignment tolerance of 500 ppm, baseline subtraction using TopHat, SavitzkyGolay smoothing (0.2 *m*/*z* width) and SNAP2 as a peak detection algorithm (S/N threshold of 3).

### 2.4. Post-MSI Staining

Following data acquisition, matrix was removed using 50% ethanol (*v*/*v*, Merck) for 5 min and H&E stained for pathological examination. Slides were scanned using a nanozoomer (Hamamatsu, China) at 43 × resolution (0.23 µm/pixel), annotated by one experienced pathologist using NPD.view 2.6.13 (Hamamatsu, China) and co- registered to MALDI MSI data using flexImaging software.

### 2.5. MALDI MSI Data Processing

Imaging datasets were uploaded and pre-processed into the SCiLS lab v2016b (Bruker Daltonik, Germany), baseline subtraction by TopHat and normalisation based on total ion count (TIC) was performed. Based on the pathological annotations, region(s) of interest from each patient core were defined and grouped together in different classes: normal and tumour for CRC while with LNM and without LNM for EC. The MSI dataset was then exported in the vendor- neutral imzML data format for supervised deep learning [20]. The mass spectrometry proteomics data have been deposited to the ProteomeXchange Consortium via the PRIDE [21] partner repository with multiple dataset identifiers, PXD019653, PXD019662, PXD019666 and PXD025594.

### 2.6. MALDI MSI Data for Machine Learning: TMAs, Patients, and Spectra

MALDI MSI typically produces hundreds of spectra spatially distributed across a sample, although tissue degradation or loss during sample preparation can result in low spectra counts. For CRC TMAs, of these 302 patients analyzed, spectra from both tumour cores and normal cores were obtained for 262 patients, tumour only spectra for 16 patients, and normal only spectra for 24 patients, resulting in a total of 564 tissue group samples. For normal samples with detected spectra, the minimum number of spectra was 13 and the maximum 535, with a median of 204; for tumour samples, the minimum number of spectra was 21 and the maximum 854, with a median of 478.5. For EC TMAs, out of 257 patients analyzed, 209 patients without LNM and 42 with LNM yielded viable data, resulting in a total of 251 tissue group samples. The minimum number of spectra per tissue group was 3 and the maximum was 175, with a median of 104.5.

### 2.7. Data Preprocessing for Machine Learning

#### 2.7.1. Assignation of Spectra to Tissue Groups

The raw data inputs for ML were available in the form of imzML (https://ms-imaging.org/wp/imzml/, accessed on 15 February 2020) files, and the publicly available Python library pyimzML (https://github.com/alexandrovteam/pyimzML, accessed on 15 February 2020) was used to extract the spectral data from the files. For CRC TMAs, the MALDI MSI data was stored in separate files for normal or tumorous samples, according to the position of the cores in the TMA design and included information on the spatial location of each spectrum. As multiple normal and tumour cores were represented for each patient, these were aggregated to define patient normal or tumour tissue groups. For EC TMAs, the data were aggregated in a similar fashion, except that patients were split in groups according to presence of absence of LNM.

#### 2.7.2. Binning of *m*/*z* Points

Each spectrum included between 171,099 and 171,178 *m*/*z* locations. The minimum *m*/*z* point varied in the interval (799.7571411132812, 800.1551513671875) while the maximum varied in (4493.67919921875, 4496.83984375).

We preprocessed the data using binning, to reduce the number of *m*/*z* points, for two reasons. First, for ML, 171,178 is an excessive number of features. Many algorithms are known to learn better if the number of input features are reduced, especially when those features are highly correlated, which is likely here between adjacent points on the *m*/*z* axis. A second reason for binning was that the data for the different TMAs had varying numbers of *m*/*z* points, and start and end *m*/*z* values, but for ML it is desirable for all samples to have the same numbers of features, and for each feature to correspond to the same physical source. The algorithm for binning was as follows:

Select a *m*/*z* bin size (we used 3, but bins of 6, 12 and 24 produced similar results).

Calculate the total number of bins, B, needed to span the range of *m*/*z* values between the minimum and maximum *m*/*z* values from all TMAs for each dataset.

Define *m*/*z* bin boundaries that are linearly spaced between the minimum and maximum *m*/*z*, such that B bins result.

For each TMA:(a)Assign each *m*/*z* point used to the nearest bin defined in step 3.(b)For each spectrum, sum the intensities over each original *m*/*z* point assigned to each bin.

When the binning process with an *m*/*z* bin size of 3 was used, the above procedure resulted in 1232 *m*/*z* bins. This meant all samples we used for machine learning had 1232 features.

#### 2.7.3. Dynamic Range Reduction

Following binning, we transformed the value in each bin in each spectrum by calculating the square root. The reason for this is that the mean value in each bin was much higher for smaller *m*/*z* values than larger ones, and reducing the dynamic range was expected to make it faster for ML algorithms to converge to stable solutions.

### 2.8. Machine Learning Methods

#### 2.8.1. Binary Classification

We aimed to algorithmically classify samples as belonging to one of two possible categories. For binary classification, we used ML, to build a model using a subset of respective training samples. The trained model was then applied to data from held out samples for classification (this later step is called ‘inference’). Two kinds of errors can occur in binary classification: false positives (FPs) and false negatives (FNs). Below, we discuss the metrics we used to examine both kinds of errors.

#### 2.8.2. Leave-One-TMA-Out Cross Validation

In any ML approach, it is essential to omit a subset of overall data from that used to train a model. This enables testing of the extent to which the model generalizes to data that has not been learned from, and to assess the extent of overfitting of the training model. For CRC TMAs, our choice for validation was to hold out from training all data from a subset of patients. This is appropriate for both per-spectra and per- tissue-group classification. For per-spectra classification, this ensures that spectra from the same patient do not appear in both the training and validation set, while for per- tissue-group classification, this ensures that tissue groups from the same patient also only get assigned to only one of the two sets. As the total number of patients was small from a ML point of view, we chose to perform cross-validation rather than fix a standard hold-out set. The advantage of cross-validation, omitting different random subsets (termed k-fold cross validation), is that a more robust model can be designed based on average performance on many validation sets.

For CRC TMAs, to account for individual TMAs possibly having distinct characteristics, we chose to carry out comprehensive Leave-One-Out cross-validation using all patients from entire individual TMAs, rather than randomly selecting patients. We call this “leave-one-TMA-out” cross validation. This choice simulates a real-world scenario where a model is trained on many TMAs, and later used to make predictions on new TMAs.

For EC TMAs, we instead chose 10-fold cross-validation with 10 randomly chosen examples from each class in each validation split. We made this choice for two reasons. First, we had only three TMAs. Second, we only had 43 samples with LNM status compared to 214 without LNM, and these were randomly spread over the three TMAs. Leaving one entire TMA out under these circumstances would have risked not including enough data from “with LNM” class in either the training or validation sets.

#### 2.8.3. Classification Thresholds

Many binary classification algorithms produce a real-valued ‘confidence score’ between zero and unity as their raw output. To produce a final classification of a sample during inference, this real number needs to have a decision threshold applied, such that if the confidence score is smaller than the threshold, the classification is the negative class, and otherwise the classification is the positive class. It should be noted that we have not attempted to set criteria for selecting a threshold for binary classification, and instead simply used the standard default value of 0.5. If this threshold is raised or lowered, it results in selecting a different point on a classifier’s receiver operating characteristic area under the curve (ROC-AUC), and a different trade-off between false negatives and false positives.

#### 2.8.4. Performance Metrics

The metrics used to assess the performance of our models are detailed in Table 1.

For model assessment, we determined the average of each of these metrics for each validation fold in our leave-one-TMA-out (for CRC) or 10-fold (for EC) cross- validation schemes.

#### 2.8.5. Samples and Model Design for Sample Classification by Spectra

For sample classification by spectra, we treated each spectrum from all TMAs as independent sample. In terms of each sample providing different information to learn from, this choice is justified by the fact that each tissue sample had spatial extent, and it can be expected that there are spatially varying signatures of the presence of a tumour. For this approach, the total number of independent samples for ML in the CRC TMA dataset was 188,515 and, in the EC TMA, dataset was 31,412. Overall, with an *m*/*z* bin size of 3, we therefore had a data matrix (prior to splitting to training and validation sets) of size 188,515 rows and 1232 columns for CRC and 31,412 row and 1232 columns for EC.

We investigated several standard ML models, including a multi-layer perceptron (i.e., a non-convolutional deep neural network), gradient boosting decision trees, and a 1-D deep convolutional neural network [22]. Our preliminary exploratory ML design work indicated that the two mentioned neural network algorithms both gave nearly the same performance, but gradient boosting decision trees performed so badly that it was not further considered. We found that training both types of neural networks and averaging their class prediction confidences prior to thresholding for classification produced the best model, and we hence present results only for this case. The neural network details are as follows.

The multilayer perceptron was a deep neural network with three layers of weights. The first hidden layer was of size 1024, and the second of size 512, both with rectified linear unit (ReLU) activations. The output layer was single dimensional with sigmoidal activation. Overall, the network had a total of 1,787,905 trainable parameters. The network was trained using binary cross entropy loss and stochastic gradient descent, with a batch size of 512, and momentum of 0.9. Training proceeded initially for 20 epochs with a learning rate of 0.01 and then for another 10 epochs with a learning rate of 0.001. To address class imbalance, a class weighting of 2.5 was applied to positive samples during training.

The 1D conv net had a total of 12 1D convolutional layers, with each except the first preceded by a batch normalization layer and then ReLU activation. The first layer was preceded only by batch normalization. Following the final convolutional layer, we used 1D global average pooling followed by sigmoidal activation. The kernel size for all 1D convolutional layers was 9. Stride of size 4 was used in the first conv layer, and stride of 3 in the 6th and 10th, with stride 1 otherwise. The number of convolutional filters were 32 (conv layer 1), 16 (conv layers 1 through 5), 32 (conv layers 6 through 10), 64 (conv layer 11) and 1 (the final conv layer). Biases were not used since they are implicit in the batch normalization layers. Overall, the network had a total of 72,802 trainable parameters. The much smaller number than the multilayer perceptron is not atypical; one of the known benefits of convolutional neural networks is their capability of learning well with many fewer parameters. The network was trained using binary cross entropy loss and stochastic gradient descent, with a batch size of 512, momentum of 0.9, and weight decay for each layer of 0.001. Training proceeded initially for 5 epochs with a learning rate of 0.1 and then for another 5 epochs with a learning rate of 0.01 and then one final epoch with a learning rate of 0.001. Class weightings were found to add no value for this model.

Both models were designed and trained using TensorFlow 2.1, following a typical rapid iteration cycle starting with typical model designs for one dimensional inputs with long locally-correlated feature vectors. Extensive hyper-parameter optimization was not attempted, since it is clear that additional data in the future will be the best path towards exceeding the performance metrics we report in this paper; significant amounts of additional data are likely to render optimized hyper-parameters based on existing data obsolete. Moreover, we used precisely the same models for both CRC and EC experiments, with the objective of developing a generally useful approach, rather than one optimized for a specific data set, or relying on one choice of machine learning algorithm.

#### 2.8.6. Samples and Model Design for Sample Classification by Tissue Group

Sample classification by tissue group required aggregation of all spectra for each of the distinct tissue groups, and fusion of the predictions made on each spectra within a group, to classify the group as normal or tumour (CRC) or according to the LNM status (EC). Our design for fusion was as follows. The raw classifier outputs as taken from the outcomes of sample classification by spectra are ‘confidence values’ that spectra are from the tumour class (a real- valued number between zero and one). The median value of the raw confidences for ‘tumour’ (calculated from all spectra for a tissue group) were thresholded to make the final classification decision.

## 3. Results and Discussion

MALDI MSI has potential to provide molecular information in the histological context [23], without the need to identify a protein of interest first and then quantify its abundance by IHC. It will just be a matter of time, until MALDI MSI is fully automated and the cost for each sample can be reduced to an acceptable level for routine analysis. MALDI MSI has already been hypothesised to be well suited for diagnostic tumour classification [23]. Several studies have shown the strength of MALDI MSI in combination with ML approaches to aid in the diagnosis and/or prognosis of different cancer types. Balluff et al. demonstrate that the MALDI MSI combined with advanced statistical clustering methods can be used to identify phenotypically and molecularly distinct tumour subpopulations with clinical significance in breast and gastric cancer [24]. Klein et al. have shown that the MALDI MSI derived proteomic features in combination with ML algorithms can classify different histologic subtypes of epithelial ovarian cancer i.e., ovarian clear-cell, high-grade serous, low-grade serous carcinomas, and serous borderline ovarian tumour [9]. Meding et al. classified six common cancer types based on the MALDI proteomic profiles [13]. The classifiers can discriminate between different tumour types at different organ sites and in the same site. However, MALDI MSI combined with ML showed great promise, it is still very new and long way to go before reaching widespread clinical use.

Cancer diagnosis and staging are essential to determining patient management. To identify robust cancer biomarkers, large numbers of patient samples must be investigated. Collections of archived formalin fixed paraffin embedded (FFPE) tissue specimens provide an excellent source for these studies. However, MALDI MSI analysis of individual tissue samples one at a time remains labor intensive and time consuming and increases the likelihood of introducing discrepancies from tissue to tissue over time. To address this issue, tissue micro arrays (TMAs) representing multiple patients can be constructed, with each patient generally represented by multiple tissue cores. Nonetheless, processing and analysis of high dimensional MALDI MSI data from large cohorts of patients remains challenging. We have previously applied a Canonical Correlation Analysis (CCA) based classification method that ranks the intensities of the *m*/*z* values to reduce the dimensions of MALDI MSI data, achieving 88% accuracy in discriminating primary carcinomas of endometrial cancer with metastatic potential (*n* = 16) from those without (*n* = 27) [16,25,26,27]. Here, we further develop our approach by applying ML to MALDI MSI data from two large patient cohorts representing different diseases, EC and CRC. Although our previous study resulted in a slightly higher classification accuracy using a smaller patient sample size, the two approaches are not directly comparable. Moreover, small sample size always carries the risk of over-fitting the model.

Adoption of supervised ML for analysis of MALDI MSI imaging data promises to strengthen current histopathological classification methods adopted by pathologists. Using supervised ML of MALDI MSI data from CRC and EC TMAs, in conjunction with pathologist annotation and patient meta data, this study set to establish approaches for binary sample classification problems.

### 3.1. Validation Results for CRC TMAs

In leave-one-TMA-out cross-validation, with models applied to individual spectra, the AUC varied between 96.8% and 99.7% for the 15 TMAs (Figure 3, inset). ROC-AUC analysis for the 15 independent classifiers (Figure 3, main axes) showed that sensitivity and specificity were both high, without a trend for one to be higher than the other, indicating that our classifiers were not biased towards one class or the other at the default threshold of 0.5.

Apart from showing that classification accuracy is high for all TMAs, an interesting finding is that there is variability in the sensitivity- specificity trade-off across TMAs. For example, TMA 6 exhibits higher specificity (97.4%) than specificity (84.4%), while TMA 13 shows the opposite pattern (93.9% sensitivity, 84.3% specificity. Such an effect in a supervised learning model can be caused by small inconsistencies between how each TMA was originally labelled, or through natural variations in the data itself, in this case either in the MALDI process or the tissue samples.

The results for sample classification by tissue group, combining the predictions of our ML model applied to individual spectra, are shown in Figure 4. Figure 5A shows histograms of the raw signal resulting from this process, prior to thresholding at a value of 0.5. Taken together these data indicate very good separability of the two classes at a threshold value of 0.5, since most tissue groups in each class have prediction probabilities close to 0 or 1, as desired. Overall, this approach achieved a mean accuracy of 98.5% over all 15 TMAs, i.e., only 1.5% of tissue groups were incorrectly classified following for sample classification by tissue group. 

The results for CRC TMA analysis highlight the capability of our supervised machine learning model to distinguish CRC tumourous tissue from the normal with over 98% accuracy. This is achieved based on the MALDI MSI data alone, without any need to identify the *m*/*z* values that produced this separability.

### 3.2. Validation results for EC

In cross-validation, with models applied to individual spectra, the AUC varied between 57% and 91% for the 10 folds as shown in Figure 5B. The ROC- AUC curves for the 10 classifiers (Figure 6, main axes) suggest that both sensitivity and specificity can be high simultaneously, as confirmed by Figure 6 (inset), but clearly some samples are difficult to classify correctly. The accuracy and balanced accuracy are identical in our setup, since each validation fold contained 10 samples from each class.

Figure 7 shows the results of applying our sample classification by group approach, combining the predictions of our ML model applied to individual spectra. Figure 5B shows the raw signal resulting from this process, prior to thresholding at a value of 0.5. The data in these figures indicates very good separability of the two classes and that 0.5 is a good value for the threshold, since most tissue groups in each class have signals close to 0 or 1, as desired. From Figure 6 we can see that the mean accuracy over all 10 folds is 79.5%. In other words, 17% of samples were incorrectly classified following use of our fusion process.

Adopting the same supervised approach for CRC to acquired MALDI MSI EC data allows classification of LNM status from MALDI MSI of EC TMAs, with 80% accuracy, when using a 10-fold cross-validation process. For the study presented here, access to supporting ground truth clinical data offered the ability to adopt a supervised convnet approach and evaluate this across two pathologies (i.e., CRC and EC).

Overall, the variability we observed in the sensitivity-specificity trade-off suggests that further improvement in the classifier design can be made to ensure the same threshold gives the same trade-off characteristic. One area to investigate would be whether normalization steps in pre-processing affect this aspect. The imzml data provided for subsequent analysis was normalized against the TIC, and as previously discussed [28], should be further evaluated to determine effects on sensitivity and specificity. Another area is whether validation variability will reduce simply by acquiring and utilizing more training data, or data augmentation. Future evaluation requires access to larger CRC and EC tissue cohort to allow this.

## 4. Conclusions and Future Work

Molecular profiling of tissue using MALDI MSI has the potential to refine traditional histopathologic classification. MALDI MSI has developed to be a powerful mass spectrometry technique that only requires standard tissue sections that can easily be obtained from routine FFPE pathology specimens. One of the major advantages of MALDI MSI over other tissue-based mass spectrometry techniques is the ability to preserve the spatial localization of biomolecules as other techniques rely on tissue homogenization. MALDI MSI provides the opportunity of assessing thousands of biomolecules at cellular level with direct correlation to histological features. Coupling MALDI MSI with a supervised ML approach, we have developed a new classification approach that can distinguish colorectal tumour from normal tissue and predict LNM status for patients with primary tumours of EC with high confidence. With the recent developments in MS, we envision that in near future, such approaches might be able to provide valuable information to pathologists and combined with ML showed great promise to help classify primary tumours. This additional capability might assist pathologists and oncologists in their therapy decision. In summary, we have successfully demonstrated that cancerous CRC tissue could be separated from the normal tissue and primary endometrial carcinomas with LNM and without LNM by using the developed ML approach.

## Figures and Tables

**Figure 1 cancers-13-05388-f001:**
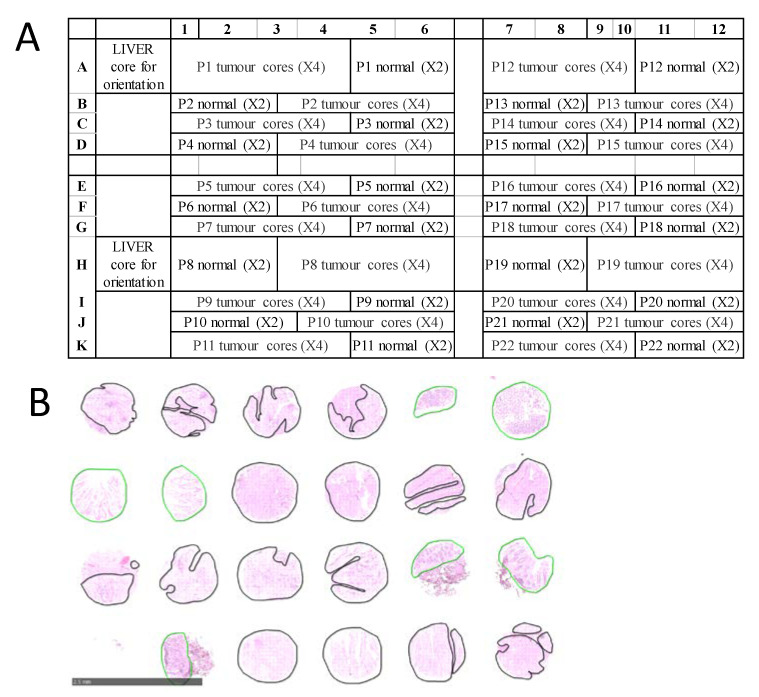
Example CRC TMA. (**A**) TMA layout, each patient has six tissue cores (two ‘normal’ and four ‘tumour’). For example, the cores for patient 1 are placed in Row A, columns 1 to 6. (**B**) Representative zoomed in annotated CRC TMA cores post MALDI MSI, with pathologist annotated regions outlined in green (normal tissue) or black (tumour tissue). Scale bar is 2.5 mm.

**Figure 2 cancers-13-05388-f002:**
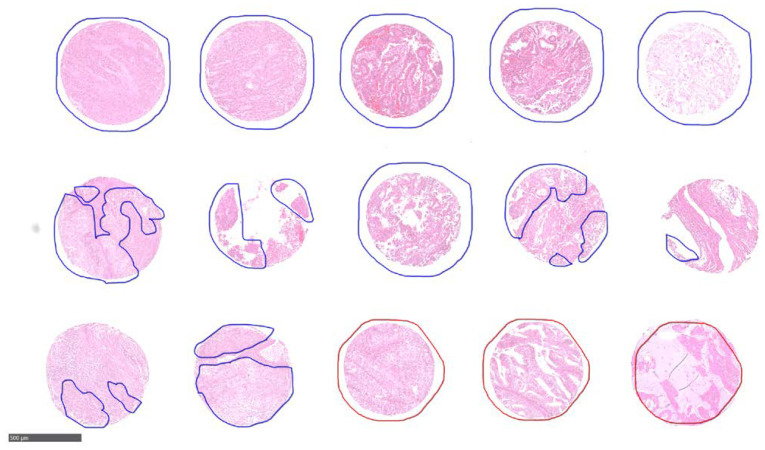
Representative zoomed in annotated EC TMA. Each patient had either two or four tissue cores, with pathologist annotated regions outlined in blue or red. Tissue cores from patients with LNM are annotated in red while blue are regions without LNM. Scale bar is 500 µm.

**Figure 3 cancers-13-05388-f003:**
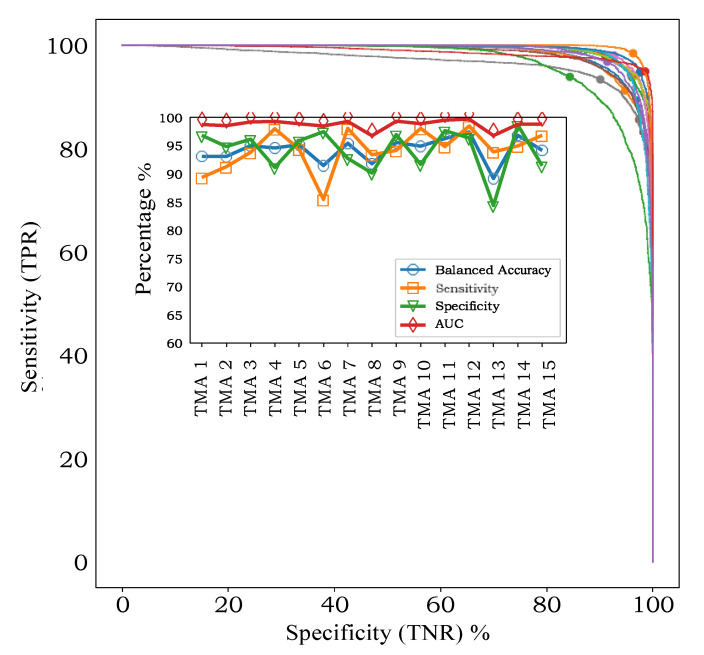
Colorectal cancer: Leave-one-TMA-out cross-validation results for classification of individual spectra. The main figure shows the ROC-AUC curves [true positive rate (TPR) vs. true negative rate (TNR)] for each TMA. The solid circle on each curve shows the operating point for a classification threshold of 0.5. The total area under the curve (AUC) for each TMA is shown in the inset, along with the corresponding accuracy, balanced accuracy, sensitivity, and specificity. The different colors of the ROC-AUC curves indicate each different TMA.

**Figure 4 cancers-13-05388-f004:**
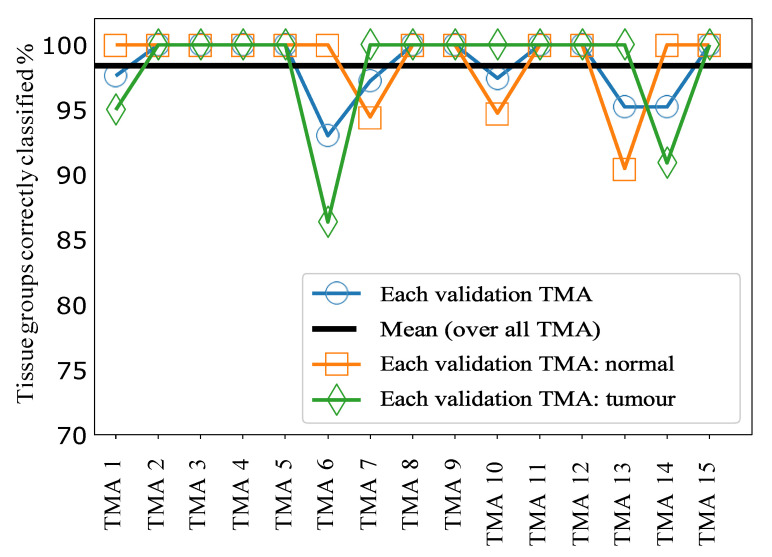
Colorectal cancer: Leave-one-TMA-out cross-validation results for Aim 2 (classify tissue groups): Each point shows the percentage of tissue groups on each TMA accurately classified, with different colors/markers indicating whether all tissue groups are included in the calculation, or only normal or tumour. The mean line is for the case of all tissue groups. The lines joining markers are a guide to the eye.

**Figure 5 cancers-13-05388-f005:**
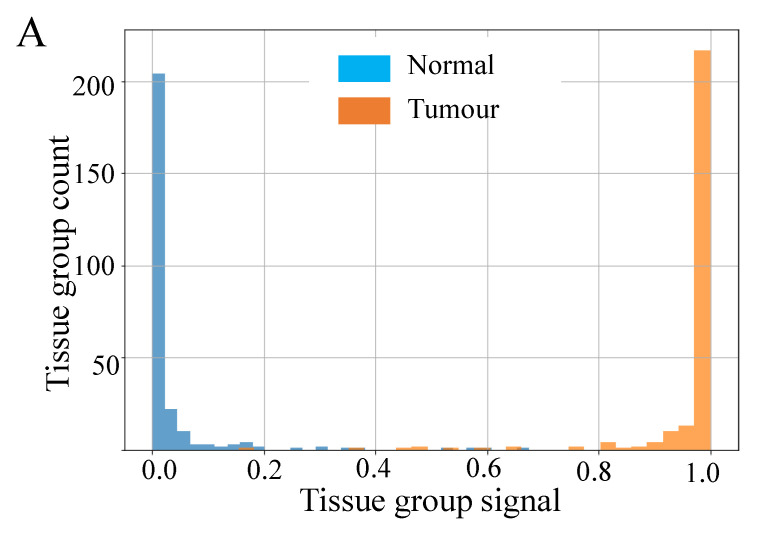
Histograms of the classifier signal produced by the final design for per-tissue-group classification for CRC (**A**) and EC (**B**). For CRC, the blue histogram is for normal tissue groups, and orange is tumour tissue groups. For EC, the data is aggregated over all 10 validation folds; the blue histogram is for LNM, and orange is for non-LNM samples.

**Figure 6 cancers-13-05388-f006:**
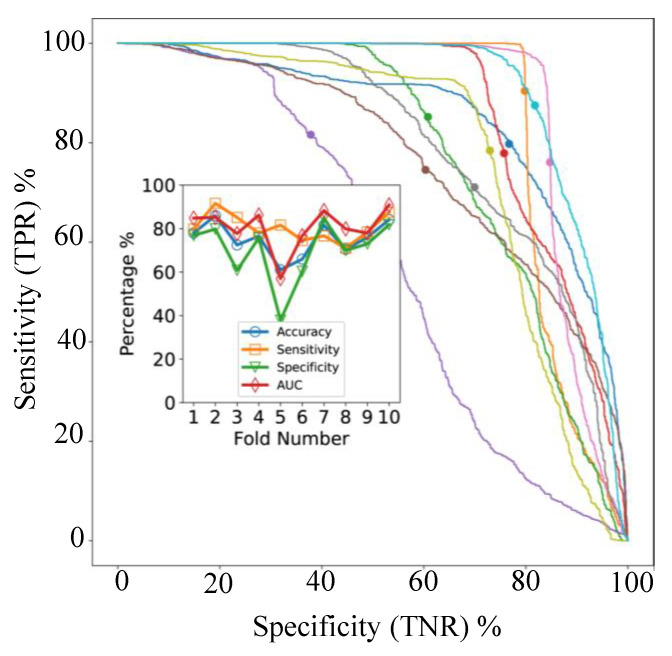
Endometrial cancer: 10-fold cross-validation results for per-spectra classification. The main figure shows the ROC-AUC curves (true positive rate (TPR) vs. true negative rate (TNR) for each fold. The solid circle on each curve shows the operating point for a classification threshold of 0.5. The total area under the curve (AUC) for each TMA is shown in the inset, along with the corresponding accuracy, sensitivity, and specificity. The different colors of the ROC-AUC curves indicate different cross-validation folds. The lines joining markers in the inset are a guide to the eye.

**Figure 7 cancers-13-05388-f007:**
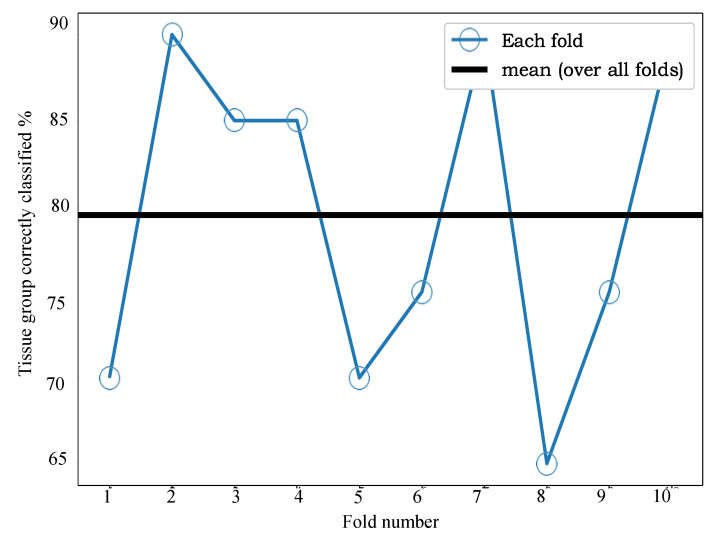
Endometrial cancer 10-fold cross-validation accuracy for per-tissue-group classification. The mean classification accuracy over the 10 folds is 79.5%. The lines joining markers are a guide to the eye.

**Table 1 cancers-13-05388-t001:** Metrics and their description.

Metric	Description
Accuracy	This is the number of currently classified samples in a validation set, divided by the total number of samples.
Balanced Accuracy	This is the average of the per-class accuracies in a validation set; per-class accuracy is one way to account for imbalance in the number of samples in each class.
Sensitivity	This is the same as true positive rate (TPR) or recall; it measures the fraction of a designated ‘positive class’ (e.g., ‘tumour’) that are correctly classified.
Specificity	This is the same as true negative rate (TNR); it measures the fraction of a designated ‘negative class’ (e.g., ‘normal’) that are correctly classified.
AUC	This is a measure of the quality of binary classifier based on the classifier’s confidence scores on the validation set; it is determined without regard to the selection of a single fixed threshold for separating classes.

## Data Availability

The mass spectrometry proteomics data have been deposited to the ProteomeXchange Consortium via the PRIDE [21] partner repository with multiple dataset identifiers, PXD019653, PXD019662, PXD019666 and PXD025594. The code used to implement the machine learning models and data preparation for these models is available at the following GitHub repository link: https://github.com/McDonnell-Lab/MALDI-UNISA-ML, accessed on 24 October 2021.

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
