# Peer review of "Cancer Tissue Classification Using Supervised Machine Learning Applied to MALDI Mass Spectrometry Imaging"

_cancers, 2021, doi:10.3390/cancers13215388_

Round 1

Reviewer 1 Report

Manuscript ID cancers-1374076: Cancer tissue classification using supervised machine learning applied to MALDI mass spectrometry imaging

This paper aims to explore the diagnostic potential of mass spectrometry-based (MALDI) imaging for tumor classification based on proteomic fingerprints. To this aim, the authors collected normal and cancer samples from 302 colorectal (CRC) and 257 endometrial cancer (EC) patients. The samples were put on tissue microarray (TMA) in which each patient is represented by multiple tissue cores. The TMA blocks were sectioned and mounted on two different types of slides for MALDI MSI and for H&E staining. The diagnosis on the H&E-stained slides was used as the ground truths for classification. The MALDI MSI data were classified by machine learning (ML) approaches. The authors found that ML-based methods successfully distinguish normal and CRC samples with an accuracy of 98% and predict the presence of lymph node metastasis for EC with an accuracy of 80%. The author concluded that MALDI is able to complement classical histopathological examination for cancer diagnostic applications.

Overall, this is an interesting paper that has a potential to solve some ambiguous histopathological cases by analyzing/classifying proteomic data using ML approaches. It has a timely contribution to the field.

General questions:

  1. The authors adopted two deep neural networks for classification, one is a multi-layer perceptron and the other is a 1-D CNN. But the architectures of the networks are not given. It is also not clear what hyper-parameters of the networks have been examined. These specifics need to be described in details.
  2. Using a limited size of EC samples, the authors achieved 88% accuracy in an earlier publication. In the present study, more EC samples were used but the prediction accuracy is decreased to 80%. It is counter-intuitive because in general more samples mean a higher prediction accuracy. The authors should discuss this result.
  3. The authors claim in the abstract that MALDI MSI has a potential to “complement” the histopathological diagnosis. It would be interesting to discuss the accuracy of ML on pathological images and compare with the accuracy of ML on MALDI MSI data to see which is higher. If MALDI MSI can really solve some histopathologically ambiguous cases in other studies, they should be mentioned or discussed in Introduction or Discussion. MALDI MSI is expensive. Does it make economic sense to replace histopathological analysis by MALDI MSI? If not, in which way MALDI MSI can “complement” the histopathological diagnosis?

Technical questions:

  1. EC TMA is two times denser than CRC TMA. Each CRC TMA has 132 tissue cores whereas on average each EC TMA has 786 / 3 = 262 tissue core. What is technical consideration behind it?
  2. All computer code for raw data processing (e.g., assignation of spectra, binning, dynamic range reduction) and machine learning needs to be well documented and stored in public repository such as GitHub. The steps need to be described in detail. For example, Pg 7, Ln 247, “The raw data inputs for ML were available in the form of imzML files, and the publicly available python library pyimzML was used to extract the spectral data from the files.” The code running the python library needs to be provided and well documented.

Author Response

  1. The authors adopted two deep neural networks for classification, one is a multi-layer perceptron and the other is a 1-D CNN. But the architectures of the networks are not given. It is also not clear what hyper-parameters of the networks have been examined. These specifics need to be described in detail.

Nearly a page of detail has been added to Section 2.8.5.

  1. Using a limited size of EC samples, the authors achieved 88% accuracy in an earlier publication. In the present study, more EC samples were used but the prediction accuracy is decreased to 80%. It is counter-intuitive because in general more samples mean a higher prediction accuracy. The authors should discuss this result.

Our previous study only contained 43 patients and was well balanced between carcinomas with and without LNM. For the classification, peak groups were generated from the MALDI data and then ranked using CCA. The data were reduced to the top m/z bins (peak groups), which can discriminate between the primary carcinomas. Using LDA to discriminate between primary carcinomas with and without LNM, a classification accuracy of 38 out of 43 patients (88.4%) was achieved by LOO cross validation. In contrast, here we didn’t reduce the data size, the patient cohort is larger and the number of carcinomas with and without LNM is not as well balanced as in our first study. To clarify this difference in approaches and comparability of results, we have included the following sentence in the discussion:

“Although our previous study resulted in a slightly higher classification accuracy using a smaller patient sample size, the two approaches are not directly comparable. Moreover, small sample size always carries the risk of over-fitting the model.”

  1. The authors claim in the abstract that MALDI MSI has a potential to “complement” the histopathological diagnosis. It would be interesting to discuss the accuracy of ML on pathological images and compare with the accuracy of ML on MALDI MSI data to see which is higher. If MALDI MSI can really solve some histopathologically ambiguous cases in other studies, they should be mentioned or discussed in Introduction or Discussion. MALDI MSI is expensive. Does it make economic sense to replace histopathological analysis by MALDI MSI? If not, in which way MALDI MSI can “complement” the histopathological diagnosis?

Following paragraph has been added in the revised manuscript

MALDI MSI has potential to provide molecular information in the histological context [1],  without the need to identify a protein of interest first and then quantify its abundance by  immunochemistry. It will just be a matter of time, until MALDI MSI is fully automated and the cost for each sample can be reduced to an acceptable level for routine analysis.

MALDI MSI has already been hypothesised to be well suited for diagnostic tumour classification [1]. Several studies have shown the strength of MALDI MSI in combination with ML approaches to aid in the diagnosis and/or prognosis of different cancer types. Balluff et al. demonstrate that the MALDI MSI combined with advanced statistical clustering methods can be used to identify phenotypically and molecularly distinct tumour subpopulations with clinical significance in breast and gastric cancer [2]. Klein et al. have shown that the MALDI MSI derived proteomic features in combination with ML algorithms can classify different histologic subtypes of epithelial ovarian cancer i.e., ovarian clear-cell, high-grade serous, low-grade serous carcinomas, and serous borderline ovarian tumour [3]. Meding et al. classified six common cancer types based on the MALDI proteomic profiles. The classifiers can discriminate between different tumour types at different organ sites and in the same site [4].

Technical questions:

  1. EC TMA is two times denser than CRC TMA. Each CRC TMA has 132 tissue cores whereas on average each EC TMA has 786 / 3 = 262 tissue core. What is technical consideration behind it?

The TMAs were received from laboratories in Melbourne, Australia and Tübingen, Germany and their standard pathology procedure, technical capability or design seems to differ between laboratories.  

  1. All computer code for raw data processing (e.g., assignation of spectra, binning, dynamic range reduction) and machine learning needs to be well documented and stored in public repository such as GitHub. The steps need to be described in detail. For example, Pg 7, Ln 247, “The raw data inputs for ML were available in the form of imzML files, and the publicly available python library pyimzML was used to extract the spectral data from the files.” The code running the python library needs to be provided and well documented.

A public GitHub repository has been created and the URL is now provided in the paper. The following text was added to the paper at the end of Section 1:

“The code we used to implement machine learning models and data preparation for these models is available at the following GitHub repository: https://github.com/McDonnell-Lab/MALDI-UNISA-ML .”

            REFERENCES

  1. Rauser, S., et al., Approaching MALDI molecular imaging for clinical proteomic research: current state and fields of application. Expert Review of Proteomics, 2010. 7(6): p. 927-941.
  2. Balluff, B., et al., De novo discovery of phenotypic intratumour heterogeneity using imaging mass spectrometry. The Journal of Pathology, 2015. 235(1): p. 3-13.
  3. Klein, O., et al., MALDI-Imaging for Classification of Epithelial Ovarian Cancer Histotypes from a Tissue Microarray Using Machine Learning Methods. PROTEOMICS – Clinical Applications, 2019. 13(1): p. 1700181.
  4. Meding, S., et al., Tumor classification of six common cancer types based on proteomic profiling by MALDI imaging. J Proteome Res, 2012. 11(3): p. 1996-2003.

Reviewer 2 Report

This study conducted by Mittal et al. attempted to develop a neural net-based machine learning (ML) model for classification of cancer tissue in colorectal- (CRC) and endometrial cancer (EC) based on MALDI MSI. The algorithms were established on TMA slides from 302 CRC and 257 EC patients. Pathologist’s annotations on HE slides was used as a gold standard. Leave-one-out and 10-fold cross validation methods were used as validation sets. The concept of developing a machine read tool that can aid the daily clinical practice of cancer diagnostics is current. Although the above-described method showed good performance in distinguishing normal - from cancer tissue, it is an easy target, and done in daily pathology practice as the first and quickest step of diagnostic process (e.g: frozen section with result of malignancy within 45 minutes). To predict lymph node status based on primary tumor only is a much more compelling aim, however the models’ performance did not reach the current clinical gold standard. Both aspects weaken the clinical significance of this study. The manuscript is easy to follow, methods are appropriate, however not well-organized. I have major comments and questions enumerated below: 1) The authors should explain the statement of purpose in more detail. In the current state, it is not clear whether the authors aimed to predict lymph node status based on the primary tumor or detect cancer metastasis in lymph node. Which one did they aim for? 2) The authors should show the detailed clinicopathological characteristics of the patient cohorts involved in the study as it can deepen the clinical context. For example, how many high grade and low grade tumors were involved, as they have very different morphology and potentially spectral distribution. Same applies for histological subtype. How was the stage distribution? Was there any pre-op treated cases involved? 3) The authors states that they performed HE staining after MSI data acquisition. How did MALD affect the histological morphology compared to a standard HE slide? 4) How many pathologists did annotate the gold standard? Did the pathologist(s) perform the annotation on post MSI HE slides or on a standard HE slide? If the morphology was affected, how the authors ensured that the gold standard is correct? 5) It would be great to see figures of tissues (Fig 1 and 2) with higher resolution, as in the current state, one cannot see the tissue type. 6) The introduction is too long and redundant with following sections. Many points mentioned in the introduction should be detailed in the discussion instead. 7) The authors states in the conclusion that “ such approached may replace the traditional way of identification, which are either time consuming or sometimes unreliable.” The authors did not investigate this aspect as an aim and did not present data to support this conclusion. Could the authors please present results or cite a previous paper to support this statement regarding their two aims (cancer vs normal tissue and lymph node metastasis)?

Author Response

  1. The authors should explain the statement of purpose in more detail. In the current state, it is not clear whether the authors aimed to predict lymph node status based on the primary tumor or detect cancer metastasis in lymph node. Which one did they aim for?

We apologise for the confusion; we have made the statement clearer in the revised version of the manuscript. The study aimed to develop a ML model that can distinguish the primary tumours of endometrial cancer correlated from lymph node metastasis form those without.

  1. The authors should show the detailed clinicopathological characteristics of the patient cohorts involved in the study as it can deepen the clinical context. For example, how many high grade and low grade tumors were involved, as they have very different morphology and potentially spectral distribution. Same applies for histological subtype. How was the stage distribution? Was there any pre-op treated cases involved?

The aim of the current study is to develop a ML model that can distinguish the CRC tumour from normal and/or primary tumour of EC with lymph node metastasis (LNM) from those without. However, we are going to extend this study for our future analysis and include more variables such as grade, stage or any other pathophysiological information as suggested by the reviewer.

For EC, patients with LNM belongs to Stage IIIC (any grade) while patients without LNM belongs to either stage I or II (any grade).

  1. The authors states that they performed HE staining after MSI data acquisition. How did MALDI affect the histological morphology compared to a standard HE slide?

MALDI MSI doesn’t affect the molecular features, in fact histology remains intact throughout the analysis. Post MALDI MSI data acquisition, the molecular features can be correlated with histological and clinical features, reported initially by Chaurand et al. [1].

  1. How many pathologists did annotate the gold standard? Did the pathologist(s) perform the annotation on post MSI HE slides or on a standard HE slide? If the morphology was affected, how the authors ensured that the gold standard is correct?

Consecutive serial sections have been used for the histological and MALDI analysis. However, Post MALDI data acquisition, the matrix was washed, and the same slide was H&E stained, annotated and co-registered to the MALDI MSI data using flexImaging software. No morphological differences were observed between the post MALDI and pre stained (consecutive glass) slide. One experienced pathologist, co-author on this manuscript, has annotated the slides.

  1. It would be great to see figures of tissues (Fig 1 and 2) with higher resolution, as in the current state, one cannot see the tissue type.

Edited as advised

  1. The introduction is too long and redundant with following sections. Many points mentioned in the introduction should be detailed in the discussion instead.

Edited as advised

  1. The authors states in the conclusion that “ such approached may replace the traditional way of identification, which are either time consuming or sometimes unreliable.” The authors did not investigate this aspect as an aim and did not present data to support this conclusion. Could the authors please present results or cite a previous paper to support this statement regarding their two aims (cancer vs normal tissue and lymph node metastasis)?

We have weakened our statement in the revised manuscript, and the following sentences has been added in the discussion

“MALDI MSI might be able to provide valuable information to pathologists and combined with ML showed great promise to help classify primary tumours. This additional capability might assist pathologists and oncologist in their therapy decision.”

            REFERENCES

  1. Chaurand, P., et al., New Developments in Profiling and Imaging of Proteins from Tissue Sections by MALDI Mass Spectrometry. Journal of Proteome Research, 2006. 5(11): p. 2889-2900.

Reviewer 3 Report

In the manuscript "Cancer tissue classification using supervised machine learning applied to MALDI mass spectrometry imaging" by Parul Mittal et al. the authors use the cutting edge technique of MALDI mass spectrometry imaging coupled with machine learning to generate a system that allows them to classify colon and endometrial samples as normal or neoplastic. The theme is interesting and relevant, with many promising applications in the future of clinical and laboratorial pathology. The paper is well written and organized. However, even though the results look interesting, there are important aspects that should merit the attention of the authors:

  1. A more comprehensive characterization of the histological samples used in the study is needed. To more accurately understand the robustness of the system, it would be important to know the histological grade of the tumours (were there only well-differentiated tumours?) and the percentage of neoplastic tissue present in each sample analyzed. Perhaps, representative sample images should be included in order to convince the reader about the diagnosis performed.

  1. The discussion section needs to be enriched with a more in-depth discussion on the potential benefits and shortcomings of their proposed approach. In line with this, they should more extensively compare the accuracy of their method with the results of similar attempts described in the literature.

  1. The authors also should clarify the meaning of the following sentence: “With the recent developments in MS, we envision that in near future, such approaches may replace the traditional way of identification, which are either time consuming or sometimes unreliable.” Why is the traditional way of identification unreliable? In line with this, the authors should also address the issue of error in pathology, discussing what is the maximum amount of error permitted to the pathologist vs. the machine learning apparatus.

Therefore, if considered relevant for publication in Cancers, the authors will have to address the aspects highlighted above before the final format of publication is achieved.

Author Response

  1. A more comprehensive characterization of the histological samples used in the study is needed. To more accurately understand the robustness of the system, it would be important to know the histological grade of the tumours (were there only well-differentiated tumours?) and the percentage of neoplastic tissue present in each sample analyzed. Perhaps, representative sample images should be included in order to convince the reader about the diagnosis performed.

The aim of the current study is to develop a ML model that can distinguish the CRC tumour from normal and/or primary tumour of EC with lymph node metastasis (LNM) from those without. However, we are going to extend this study for our future analysis and include more variables such as grade, stage or any other pathophysiological information as suggested by the reviewer.

  1. The discussion section needs to be enriched with a more in-depth discussion on the potential benefits and shortcomings of their proposed approach. In line with this, they should more extensively compare the accuracy of their method with the results of similar attempts described in the literature.

As suggested, following paragraph has been added in the revised manuscript:

MALDI MSI has potential to provide molecular information in the histological context [1], without the need to identify a protein of interest first and then quantify its abundance by immunochemistry. It will just be a matter of time, until MALDI MSI is fully automated and the cost for each sample can be reduced to an acceptable level for routine analysis. MALDI MSI has already been hypothesised to be well suited for diagnostic tumour classification [1]. Several studies have shown the strength of MALDI MSI in combination with ML approaches to aid in the diagnosis and/or prognosis of different cancer types. Balluff et al. demonstrate that the MALDI MSI combined with advanced statistical clustering methods can be used to identify phenotypically and molecularly distinct tumour subpopulations with clinical significance in breast and gastric cancer [2]. Klein et al. have shown that the MALDI MSI derived proteomic features in combination with ML algorithms can classify different histologic subtypes of epithelial ovarian cancer i.e., ovarian clear-cell, high-grade serous, low-grade serous carcinomas, and serous borderline ovarian tumour [3]. Meding et al. classified six common cancer types based on the MALDI proteomic profiles. The classifiers can discriminate between different tumour types at different organ sites and in the same site [4]. However, MALDI MSI combined with ML showed great promise, it is still very new and long way to go before reaching widespread clinical use.

  1. The authors also should clarify the meaning of the following sentence: “With the recent developments in MS, we envision that in near future, such approaches may replace the traditional way of identification, which are either time consuming or sometimes unreliable.” Why is the traditional way of identification unreliable? In line with this, the authors should also address the issue of error in pathology, discussing what is the maximum amount of error permitted to the pathologist vs. the machine learning apparatus.

We have weakened our statement in the revised manuscript, and the following sentences has been added:

“MALDI MSI might be able to provide valuable information to pathologists and combined with ML showed great promise to help classify primary tumours. This additional capability might assist pathologists and oncologist in their therapy decision.”

            REFERENCES

  1. Rauser, S., et al., Approaching MALDI molecular imaging for clinical proteomic research: current state and fields of application. Expert Review of Proteomics, 2010. 7(6): p. 927-941.
  2. Balluff, B., et al., De novo discovery of phenotypic intratumour heterogeneity using imaging mass spectrometry. The Journal of Pathology, 2015. 235(1): p. 3-13.
  3. Klein, O., et al., MALDI-Imaging for Classification of Epithelial Ovarian Cancer Histotypes from a Tissue Microarray Using Machine Learning Methods. PROTEOMICS – Clinical Applications, 2019. 13(1): p. 1700181.
  4. Meding, S., et al., Tumor classification of six common cancer types based on proteomic profiling by MALDI imaging. J Proteome Res, 2012. 11(3): p. 1996-2003.

Round 2

Reviewer 2 Report

The authors responded to my questions, no further comments or questions.

Reviewer 3 Report

In the revised version of the manuscript "Cancer tissue classification using supervised machine learning applied to MALDI mass spectrometry imaging" by Parul Mittal et al. the authors use the cutting edge technique of MALDI mass spectrometry imaging coupled with machine learning to generate a system that allows them to classify colon and endometrial samples as normal or neoplastic. The theme is interesting and relevant, with many promising applications in the future of clinical and laboratorial pathology. The paper is well written and organized. In the revised version, the authors addressed in a satisfactory manner all my previous concerns. Therefore, I strongly recommend the publication of the paper in the current format.